# An Optimal Allocation Model for Large Complex Water Resources System Considering Water supply and Ecological Needs

**Yaogeng Tan** [1,*], **Zengchuan Dong** [1], **Chuansheng Xiong** [1], **Zhiyu Zhong** [2] **and Lina Hou** [2]

1    College of Hydrology and water resources, Hohai University; Nanjing 210098, China;
     zcdong@hhu.edu.cn (Z.D.); xcs123@163.com (C.X.)
2    Institute of water resources planning, Changjiang Institute of Survey, Planning, Design and Research,
     Wuhan 430010, China; zhongzhiyu@cjwsjy.com.cn (Z.Z.); hln20110801@163.com (L.H.)
*    Correspondence: fred_2776@sina.com

**Abstract:** Water resources are very important to support the socio-economic development and maintain environmental health, which is a typical issue in water resources management. In this study, we developed an optimal allocation model for a large complex system of water resources by considering both water supply and river ecological benefits. The water supply benefit is defined as the minimum water deficit for different water users, while the ecological benefit involves making the reservoir release as close as possible to the natural streamflow. To solve this problem, the combination of decomposition-coordination (DC) and discrete differential dynamic programming (DDDP) methods were proposed. The proposed methods first decomposed a large system with multi-objective programming into subsystems, and the optimal solution of each subsystem was accomplished by the DDDP method to solve the system efficiently. Then the subsystems' solutions were coordinated to figure out the near global optimal solution. The proposed models were tested in the Lingui and Yongfu County, Guilin City in China. Results show that the optimal reservoir release is close to the natural flow regime and there is a slight water deficit ratio in both level years. The water supply objective is more sensitive to the system model compared with the ecological objective, and the result of water allocation is optimized when the reservoir release is as close as possible to the natural flow based on the minimum water deficit. The proposed system model could facilitate sustainable water use and provide technical support for water resources management in economic development.

**Keywords:** optimal water allocation model; large complex system; system analysis technique; decomposition-coordination method; discrete differential dynamic programming; sustainable water use

## 1. Introduction

It is widely known that the continuous increase of the water demand and the limited quantity of water resources are severely restricting the development of the global economy. Constructing reservoirs is one of the crucial manifestations in solving water resources problems. However, socio-economic water supplies from reservoirs are not only changing the natural flow, but also deteriorating the riverine ecosystems by affecting the sediments' nutrient distribution and degrading the flows of the river downstream [1,2]. To reduce the negative impact of reservoirs on ecosystems, a water resource allocation approach is needed that can ensure that the water supply and the protection of the downstream riverine ecosystem can be incorporated into the management of water resources to achieve the goal of sustainable water use [3]. It is, therefore, of great significance to incorporate the needs of human society and ecological requirements.

Several mathematic programming methods have the ability to satisfy multiple water users in water resource allocation. For example, linear programming is frequently applied in water resources systems due to its easy formulation and simple arithmetic [4]. To resolve the nonlinear programming, several improved intelligent optimal algorithms have emerged as computer techniques have rapidly been developed [5,6], such as non-dominated-sorting genetic algorithm II (NSGA-II) [7,8]. Dynamic programming that uses the two-stage method is one of the most popular optimization techniques for water resources issues, such as two-stage stochastic programming (TSP) [9,10], and the progressive optimality algorithm (POA) [11,12]. These searching methods have been popular among many researchers because of their high efficiency, and they are widely used in water resources management in many regions.

Optimization technologies have been applied to solve the problem of both single-objective and multi-objective optimizations in large complex water resources systems. However, recent studies have focused more on multi-objective programming rather than single-objective programming, because it has to face different aspects of operational targets [13]. Among all objectives, ecological objectives are of great importance in water resources management. It is necessary for humans to exploit natural resources, especially water resources, in order to meet social and economic needs, but the excessive exploitation of water resources will inevitably affect the ecological environment [14–17]. In addition, humans usually take water from rivers that can no longer be considered solely to be natural systems, having rather the double property of being both natural and the result of human-hydrological interaction, resulting in an inexorable trend of social development in recent years [18]. The artificial intervention in the natural flow regime has become a key factor of severe deterioration of riverine ecosystems [19]. Therefore, as described at the beginning, environmental factors are being brought to the forefront in reservoir operation [20,21], and optimal models of water resources have considered environmental targets in many studies [22,23], increasing their integrity and practical applicability. Based on these, studying the development and utilization of water resources by using an integrated and comprehensive method is becoming an inevitable trend [24].

Comprehensive water resources planning for a watershed or region is a large and complicated system in most cases, because it involves how to reasonably utilize an integrated water resources system composed of numerous rivers in a certain region, as well as multiple development goals, restraints and effects [25]. The complexity of water resources systems is reflected in the scales of both time and space, which constitute a high-dimensioned system. The length of the time is usually divided into multiple time steps that depend on different situations. For example, to analyze the water supply, especially in agricultural settings, the time scale is usually divided into monthly time steps. If flood control during flood seasons aquatic ecology in consideration of flow velocity are studied, the time scale is usually divided into daily or even hourly time steps. However, it is too difficult to collect such high-accuracy data. The complexity in the space scale is reflected by comprehensive water resources planning for a certain river basin or region that is usually divided into several sub-areas or sub-river basins, instead of carrying out planning for the whole area. A sub-area may contain many river channels and water projects. The precision of the sub-areas also depends on the situation. Moreover, water supply is usually divided into several users such as domestic, industrial and agricultural users, and this naturally involves an order or priority among the different water users. The priority should be relatively balanced [26], but domestic water users should have a higher priority compared with others [4]. Therefore, time and space scales, coupled with a large number of water users, constitutes a complex water system with high dimensionality.

Due to this high level of complexity, there has also been an upsurge in the number of system analysis methods in recent years, such as aggregation-decomposition (AD) and decomposition-coordination (DC) in water resources management, which are appealing because of their clear hierarchical structure and organic integrated wholeness. For AD, the aggregation method regards the whole system as an integer in accordance with the water balance equation, while the decomposition method allocates water resources to each subsystem to maximize the profit [27]. To effectively reduce the system dimensions,

the DC method has also been popular in many studies [28,29]. The decomposition-coordination method of large complex system theory divides a large complex system into its smaller constituent subsystems [30]. For example, the Dantzig-Wolfe decomposition technique is a typical method for linear programming using a diagonal structure [31]; Generalized Bender Decomposition (GBD) is a powerful approach for complex non-linear programming [32]. Its main function is to reduce the system dimensionality and divide it into several hierarchical subsystems with lower dimensionality. This method requires less computing time and is significantly simplified with respect to optimizing each subsystem, because of its lower number of system dimensions. In general, if an optimal solution for each subsystem can be figured out, this will result in the optimal solution for the overall system. Meanwhile, the coordination of each subsystem can also reflect the overall optimal solution of the whole complex system [33]. Previous studies have typically placed more focus on social objectives, such as complex water supply systems [34], power generation [35,36], flood control [30], and even irrigation water use [37] by using system analysis techniques. However, for the sustainable development of a river basin or a region, the AD method is seldom used, since it is not able to entirely lead to a benefit for all parties, especially with respect to ecological needs, and water allocation should respond not only to the pursuit of socioeconomic benefit, such as achieving maximum water supply, but should also provide the maximum benefit for the environmental protection of a river in order to avoid excessive economic tendency decisions.

It is evident that socio-economic and ecological benefit are mutually contradictory. Achieving a balanced status between these two aspects to promote sustainable development in a region is, therefore, a key issue. To achieve this goal, we developed an optimal water allocation model considering social and ecological needs, and these two objectives were embedded in each subsystem by using the system analysis technique. The model was based on a system analysis technique that used the decomposition-coordination method and the discrete differential dynamic programming (DDDP) method. Based on these two methods, these two objectives can be well-coordinated and we can effectively find the optimal solution of both subsystem and overall system. The whole model was applied to a real-world simulation to validate its feasibility and to provide a reference for water resources management.

## 2. Model Construction for Water Resources Allocation

### 2.1. Water Supply Systems of Reservoirs

The relative sufficiency of the total amount is a basic characteristic of water resources in many areas, but its spatial distribution is highly uneven. Therefore, the construction of reservoirs is one of the main measures used to solve the problem of the optimal allocation of water resources. Reservoirs should maintain a certain natural flow to meet the ecological needs of water and to minimize the change of the natural streamflow [38]. Maximizing overall benefit balances the two aim in the optimal allocation of water resources.

The proposed model is based on a water supply layout that is typical for most areas with multiple reservoirs. In such areas, large reservoirs are often constructed, and these are either mixed-connected (Figure 1a)—that is, they are used in a large river basin—or they are in parallel (Figure 1b)—that is, they are usually applied only in a relatively small region [27]. This paper mainly discusses the latter scenario. Parallel reservoirs are normally individual units, and each big reservoir supplies water for a certain part of the region (Figure 1c). In addition, the water supply area is complicated, because it has many rivers with complex relationships, and a great number of water users. An area is often separated into several sub-areas, each of which manifests a complex relationship with the upstream and downstream of the river and its tributaries and streams, rather than being mutually independent. Large reservoirs supply water for a certain area, which is composed of sub-areas that can be regarded as an integer, thus comprising a large complex system of water resources (Figure 1d).

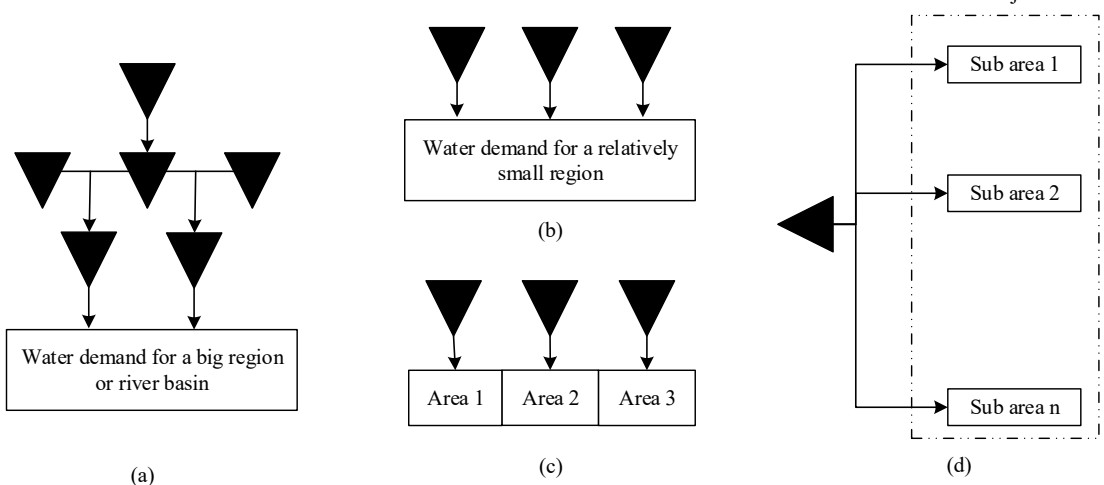

**Figure 1.** Reservoir topology structure. (**a**) Mixed-connected; (**b**) parallel; (**c**) based on (**b**), the small region divided into several areas; (**d**) the topology structure among sub-areas and reservoir.

The water resources system is a large organizational system that is mainly comprised of a large reservoir, sub-areas, water supply channels and relationships among sub-areas. A reservoir usually supplies water to more than one sub-area by changing the natural flow. Sub-areas receive water from reservoirs and from upstream, and the rest of the water will flow on to the next sub-area(s). The different sub-areas form network relations. In this study, reservoirs and their sub-areas were all considered to be subsystems. A system analysis technique was used to find the optimal solution for each subsystem under different types of objectives and constraints. The system structure of water resources and their mutual relationships are shown in Figures A1 and A2 in Appendix A, and their parameters are listed in Tables A1 and A2 in Appendix A.

### 2.2. Objective Function

The proposed model is a multi-objective optimization problem with respect to sustainable water use in terms of both water supply and riverine ecological benefit [20]. The water supply benefit aims to maximize the total water supply and to minimize the total water deficit in each area. In other words, adequate water supply is one of the most critical assurances for socio-economic development in this study. In terms of riverine ecological benefit, the alteration of the natural flow has become a crucial factor leading to the deterioration of riverine ecosystems [39]. One of the best ways to protect the riverine ecosystem is to reduce the hydrological alteration of natural flow [20,38,40]; therefore, maximizing riverine ecological benefit involves minimizing the deviation between the natural flow and the actual reservoir release downstream of the reservoir. The objective is as follows:

$$\min F_1 = \frac{1}{T}\sum_{t=1}^{T}\sum_{j=1}^{J}\sum_{k=2}^{K} TWDR_{jkt}^2 = \frac{1}{T}\sum_{t=1}^{T}\sum_{j=1}^{J}\sum_{k=2}^{K}\left(\frac{TWD_{jkt} - TWS_{jkt}}{TWD_{jkt}}\right)^2 \tag{1}$$

$$\min F_2 = \frac{AAPFD}{5} = \frac{1}{5Y}\sum_{y=1}^{Y}\sqrt{\sum_{m=1}^{12}\left(\frac{Q_{my} - QN_{my}}{\overline{Q_y}}\right)^2} \tag{2}$$

where $F_1$ and $F_2$, respectively, represent the water supply and riverine ecological objectives. The subscript $j$ and $k$ represent water users and sub-areas, respectively. Subscript $y$ and $m$ in Equation (2) represent the year and month numbers, respectively. *TWS* and *TWD* are the total water supply and water demand, respectively, and *TWDR* is the total water deficit ratio. $Q$ and $QN$ are the actual reservoir release and natural flow, respectively. $\overline{Q}$ is the average annual natural flow. The parameter

*AAPFD* is used to reflect the deviation between ecological reservoir release and the natural flow [20,41]. Gehrke [42] argued that annual proportional flow deviation (APFD) was related to the diversity of fish species, but the amended method, *AAPFD*, is proved to be better to be applied in this study using Equation (2). Actually, *AAPFD* can quantify the degree of deviation of the natural flow, and it is sensitive to modification of natural flow. Ladson [41] also found that the condition of the river ecosystem is negatively correlated with the value of *AAPFD*. The smaller the value of *AAPFD*, the smaller the alteration of the natural flow regime. Greater values of *AAPFD* signify more serious cases of hydrological alteration. When *AAPFD* is greater than five, the river ecosystem will be seriously damaged. It should be stressed that the objectives $F_1$ and $F_2$ were normalized and transformed into non-dimensioned variables ranging from 0 to 1.

### 2.3. Constraints

(1) Water balance of reservoir:

$$V_t^0 + QN_t - \sum_{j=1}^{J} \sum_{k=2}^{K} WS_{jkt} - WEVP_t - Q_t = V_t^1 \tag{3}$$

$$V_t^1 = V_{t+1}^0 \tag{4}$$

where $t$ is the time step. *WS* is the water supply from the reservoir to the sub-area(s), and *WEVP* is the surface water evaporation of the reservoir. Superscript 0 and 1 are the actual initial and terminal storages of the reservoir.

(2) The order for water supply:

The local water for the sub-areas, including the water intake from rivers and small reservoirs, is of higher priority than the large reservoirs for all sub-areas. Domestic water supply has higher priority than that for industrial and agricultural purposes [4], and the relative priority of these groups of water users should be balanced [26].

(3) Reservoir volume:

$$V_{\min} \leq V_t \leq V_{\max} \tag{5}$$

where $V_{\min}$ and $V_{\max}$ are the dead storage and useful storage of the reservoir, respectively. $V_t$ is the current storage of the reservoir.

(4) Basic ecological demand:

The basic ecological demand ensures the fundamental function of rivers. The actual release of the reservoir must maintain a certain proportion [21]. This constraint can be formulated as follows:

$$Q_t \geq QB_t, \ QB_t = QN_t \xi_t \tag{6}$$

where $QB_t$ is the basic ecological demand of the downstream reservoir, $\xi_t$ is the minimum proportion of the ecological demand in the time period $t$.

It should be noted that basic ecological demand is included in actual reservoir release, and if its value is below $QB$, the river will suffer irreversible damage.

(5) Water demand [4,35]:

$$0 \leq TWS_{jkt} \leq TWD_{jkt} \tag{7}$$

This constraint denotes that water supply should be less than water demand based on the principle of saving of resources and their effective utilization. Water supply is based on the water demand in a region. If more water is supplied, it will cause wastage of water.

(6) Water supply capacity:

$$\sum_{j=1}^{J} TWS_{njt} \leq TWS_{n\max} \tag{8}$$

where $TWS_{nj}$ is the $j^{th}$ sector's total water supply for the $n^{th}$ water supply project.

(7) Water balance of sub-areas:

$$W_{kt}^0 + I_{kt} - \sum_{j=1}^J TWS_{jkt} - WR_{kt} = W_{kt}^1 \tag{9}$$

$$W_{kt}^1 = W_{k,t+1}^0 \tag{10}$$

where $I_{kt}$ is the total water coming into the sub-area(s). $WR$ is the water recession of the $k^{th}$ sub-area. $W_{kt}^0$ and $W_{kt}^1$ are the initial and terminal total available water quantity.

(8) Interconnection among the sub-areas and the reservoir:

$$I_{kt} = \sum_{j=1}^J WS_{jkt} + \sum_{k\in\Omega} WR_{k-1,t} + WIF_{kt} \tag{11}$$

where $WIF$ is the intermediate water flow between the $(k-1)^{th}$ and $k^{th}$ sub-areas. $\Omega$ is the sum of the direct upper reaches of the subsystem $k$.

## 3. System Analysis Technique for Finding the Optimal Solution

### 3.1. Decomposition-Coordination (DC) Method

#### 3.1.1. System Decomposition

Obviously, it is quite difficult to optimize such a large and complex system using traditional methods, because of its high number of dimensions and its complicated structure. Thus, we can use the system analysis technique. The decomposition-coordination method is one of the most effective ways of implementing the system analysis technique. According to its basic principle, the key procedure of this method is disaggregating a large system into several subsystems with a hierarchical structure [33,35]. The subsystems have their own objectives, and their locally optimal solutions can be figured out.

According to the system model constructed above, the total objective and constraints of the large complex system of water resources can be expressed by the Lagrange function [29], formulated by:

$$
\begin{aligned}
\min L = F_1 + F_2 + \sum_{k=2}^K \sum_{t=1}^T \lambda_{k-1,t}\left( I_{kt} - \sum_{j=1}^J WS_{jkt} - \sum_{k\in\Omega} WR_{k-1,t} - WIF_t \right) \\
+ \sum_{t=1}^T \mu_{1t}\left( QN_t - Q_t + V_t^0 - V_t^1 - \sum_{j=1}^J \sum_{k=2}^K WS_{jkt} \right) + \sum_{k=2}^K \sum_{t=1}^T \mu_{kt}\left( I_{kt} - \sum_{j=1}^J TWS_{jkt} + W_{kt}^0 - W_{kt}^1 - WR_{kt} \right)
\end{aligned} \tag{12}
$$

where $L$ is the expression of the Lagrange function. The variables $\lambda_{kt}$ and $\mu_{kt}$ are the Lagrange multipliers, which are also called coordinative variables. Equation (12) is the expression of the Lagrange function. Meanwhile, Equation (12) can be separated as follows to make the function additive:

$$
\begin{aligned}
\min L = \left[ \sum_{t=1}^T \left( F_{2,t} - \sum_{j=1}^J \sum_{k=2}^K \lambda_{k-1,t} WS_{jkt} \right) \right] + \left[ \sum_{k=2}^K \sum_{t=1}^T \left( F_{1,kt} + \lambda_{k-1,t}\left( I_{kt} - \sum_{k\in\Omega} WR_{k-1,t} - WIF_t \right) \right) \right] \\
+ \sum_{t=1}^T \mu_{1t}\left( QN_t - Q_t + V_t^0 - V_t^1 - \sum_{j=1}^J \sum_{k=2}^K WS_{jkt} \right) + \sum_{k=2}^K \sum_{t=1}^T \mu_{kt}\left( I_{kt} - \sum_{j=1}^J TWS_{jkt} + W_{kt}^0 - W_{kt}^1 - WR_{kt} \right)
\end{aligned} \tag{13}
$$

where $F_{1,kt}$ is the water supply objective and $F_1 = \frac{1}{T} \sum_{k=2}^K \sum_{t=1}^T F_{1,kt}$. $F_{2,t}$ is the ecological objective and $F_2 = \frac{1}{T} \sum_{t=1}^T F_{2,t}$. The water balance equation is satisfied in this model; thus, the last two items of Equation (13) are equal to 0. Then, Equation (13) is defined as being the total system model Lagrange function, and can be regarded as a summary of all subsystems. In absolute terms, the equation is also

the sum of the expression of the objectives for each subsystem. In view of this, the problem of a large complex problem can be decomposed into several sub-problems, expressed as follows:

Objective:

$$minL_k = \begin{cases} \sum_{t=1}^{T}\left(F_{2,t} - \sum_{j=1}^{J}\sum_{k=2}^{K}\lambda_{k-1,t}WS_{jkt}\right), & k = 1 \\ \sum_{t=1}^{T}\left[F_{1,kt} + \lambda_{k-1,t}\left(I_{kt} - \sum_{k\in\Omega}WR_{k-1,t} - WIF_t\right)\right], & k = 2,3,...,K \end{cases} \tag{14}$$

Constraints: see Equations (3)–(11).

By converting the Lagrange function of the large system, the large-scale water resources allocation problem was divided into relatively small sub-problems. Obviously, the different objectives are reflected by decomposed subproblems. In other words, the objective function of subsystem 1 is the ecological target of the downstream reservoir, and the objective functions of subsystem 2 to subsystem *K* are the water supply targets for each sub-area. From an overall perspective, the whole model considers an ecological as well as a water supply objective. In this way, if we determine the optimal solution for all small subsystems, the solution of a large system will naturally be solved through the coordination method.

### 3.1.2. System Coordination

Figure 2 depicts the structure for both decomposition and coordination levels. Based on the optimal solution of each subsystem (Level 1 in Figure 2), the coordination process is carried out through the Lagrange multiplier $\lambda_{kt}$ and $\mu_{kt}$ to obtain the overall optimal solution. The decomposition and coordination processes are closely related. According to the dual theory, if we consider the Lagrange multiplier as variables, Equation (12) can be translated into a dual function, the maximum value of which can be determined, formulated as follows:

$$maxL = L(\lambda) \tag{15}$$

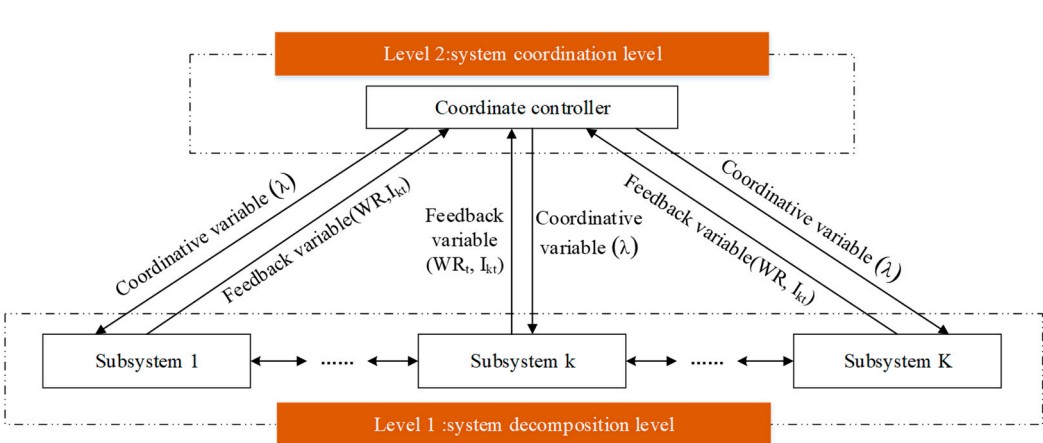

**Figure 2.** Structure of the large system based on the decomposition and coordination method.

According to the condition of obtaining the extremum value of the function, Equation (16) obtains the maximum value when the following condition is satisfied.

$$\frac{\partial L}{\partial \lambda_{kt}} = 0 \tag{16}$$

$$I_{kt}^{m} = \sum_{j=1}^{J} WS_{jkt}^{m} + \sum_{k\in\Omega} WR_{k-1,t}^{m} + WIF_t \tag{17}$$

where $m$ denotes the iteration number. Equation (17) suggests that the optimal outflow of the upper reaches of the $k^{th}$ subsystem is regarded as the inflow of the $k^{th}$ subsystem, as well as the new correlation estimate value and constraints. At this level, $WR_{kt}$ and $I_{kt}$ are feedback variables, and they are optimized in each subsystem and used to respond to the coordination level. The following condition should also be satisfied if the Lagrange function achieves its maximum.

$$\begin{cases} \dfrac{\partial L}{\partial I_{k+1,t}} = \lambda_{kt} + \mu_{k+1,t} = 0 \\ \dfrac{\partial L}{\partial WR_{kt}} = -2\left(1 - \dfrac{TWS_{kt}}{TWD_{kt}}\right) - \lambda_{kt} - \mu_{kt} = 0 \end{cases} \tag{18}$$

Then

$$\lambda_{kt} = \lambda_{k+1,t} + 2\left(1 - \frac{TWS_{k+1,t}}{TWD_{k+1,t}}\right) \tag{19}$$

$$\lambda_{kt}^{m+1} = 2\sum_{n=k+1}^{K}\left(1 - \frac{TWS_{nt}^{m}}{TWD_{nt}}\right) \tag{20}$$

The iteration equation of coordination factor $\lambda$ is obtained by Equation (20).

## 3.2. Subsystem Solving

If we take each time step as a stage, water allocation can be seen as the sequential allocation at each stage. The total input in a water resources system is usually the historical reservoir inflow over multiple years, which is always given from hydrological statistical yearbooks. Therefore, dynamic programming was formulated to effectively utilize the sequential decision structure of optimization problems [43]. In addition, this system model satisfies the condition of decision segmentation, with the mathematical model being as presented below:

(1) Stage variables: Select each time step as a stage variable, with $t = 1, 2, \ldots, T$.

(2) State variables: the initial storage of the reservoir and the initial total water quantity of each sub-area can be selected as state variables, defined by $V_{0,t}$ and $W_{0,t}$.

(3) Decision variables: $WS$ and $Q$ determine the reservoir storage, while $TWS$ and $WR$ determine the total water quantity of sub-area. It should be stressed that the value of $WS$ is determined by the value of $Q$, because of the constraint of water balance of reservoir and its storage, and $WR$ has little effect on the accuracy of the model. Therefore, $Q$ and $TWS$ are the main decision variables in this model.

(4) State transition equation: water balance equation of reservoir and sub-areas can be selected as the state transition equation, expressed by Equation (3) and Equation (9)

(5) Recurrence formulation:

$$f_t(V_t) = \min\{g_t(V_t, Q_t) + f_{t+1}(V_{t+1})\} \quad (k = 1) \tag{21}$$

$$f_t^*(W_t) = \min\{g_t^*(W_t, TWS_t) + f_{t+1}^*(W_{t+1})\} \quad (k = 2, 3, ..., K) \tag{22}$$

where $f_t(V_t)$ and $f_t^*(W_t)$ are the minimum values of an objective function of subsystem 1 and subsystem $k$ from the beginning of the $t^{th}$ time step to the $T^{th}$ time step, which can indirectly reflect the ecological and water deficit ratio. $f_{t+1}(V_{t+1})$ and $f_{t+1}^*(W_{t+1})$ are the minimum values of an objective function of subsystem 1 and subsystem $k$ for the rest period (from $(t+1)^{th}$ time step to $T^{th}$ time step). $g_t(V_t, Q_t)$ is the minimum ecological water deficit ratio of stage $t$ with the decision variables $Q_t$ at the state $V_t$. $g_t^*(W_t, TWS_t)$ is the minimum water deficit ratio of stage $t$ with the decision variable $TWS_t$ at state $W_t$.

However, when the dimension is too high, it may be impossible to figure out the optimal solution, and this is called the "curse of dimension". Therefore, an improved DP called Discrete Differential Dynamic Programming (DDDP) was used to solve this problem [44]. To improve the normal DP method, DDDP is an iteration arithmetic which gradually approximates the predetermined objective.

When seeking the optimal solution to a problem, upper and lower boundaries are needed in each stage to generate a corridor in the overall situation. The width of the corridor plays an important role in seeking the optimal results. If the corridor is too wide, it may give lower precision. Otherwise, it may introduce more complexity. The normal DP is used in the corridor to seek the optimal values. The results are not necessarily optimal at the beginning, and only change the allocation of water resources. The corridor width is usually set to be large, and it will gradually converge to a small value for precision as further iterations are executed. The optimal solution will be determined as the corridor width narrows.

### 3.3. Whole Solving Procedures

Figure 3 shows the whole procedure for the solution of the large water resources system using the DC and DDDP methods. The DC method was first used to decompose the complex large water resources system into small subsystems and to generate the corresponding sub-problems, and then all of the subsystems were coordinated to figure out their coordinative variables; this is essential to determining the optimal solution to the complex large system. The DDDP method was used to determine the optimal solution for both the subsystems and the whole system. The detailed solution procedure combining the DC and DDDP methods was as follows:

Step 1: Start the whole program.

Step 2: Initialize the parameters and set $n = 1$, where $n$ is the iteration number of the whole system. In this case, the parameters include the initial storage of the reservoir, the initial value of water quantity for each sub-area, and the water recession coefficient of this sub-area from previous sub-area(s).

Step 3: Begin calculation. Calculate the ecological release of the reservoir ($M_{1t}$) and the total water supply of each sub-area ($M_{kt}$). Reservoir ecological release can be calculated using the Tennant method, and total water supply can be calculated based on the water balance equation. In this step, the initial allocation scheme is generated.

Step 4: Calculate the coordinative variable based on Equation (20). The calculated value is set as the initial coordinative variable.

Step 5: Begin the main method. In this step, the large system is divided into subsystems, and each subsystem is optimized by DDDP; set $m = 1$, where $m$ is the iteration number of the $k^{th}$ subsystem. The optimization process is within the given width of the corridor by normal DP using the inverse sequence method to obtain the new allocation scheme and narrow the corridor. If the error-adjacent iteration of the allocation is less than $\varepsilon$, then go to the next step; otherwise, repeat this step, and set $m = m + 1$.

Step 6: Calculate the new coordinative variable and compare it with the initial one. If the convergence condition is satisfied or the maximum iteration is attained, then go to the next step; otherwise, update the coordinative variable and return to step 4.

Step 7: Prepare to optimize the next subsystem using the same method. If $k = k$max, go to the next step; otherwise, go back to step 5 to optimize the next subsystem.

Step 8: In this step, the whole optimization is finished and it is time to prepare the next iteration of the whole system. If $n = n$max or the convergence condition is satisfied, go to the next step; otherwise, go back to step 3 for the next iteration.

Step 9: The total optimal allocation scheme is generated, including the ecological release of the reservoir and socio-economic water allocation for each sub-area.

Step 10: End the program.

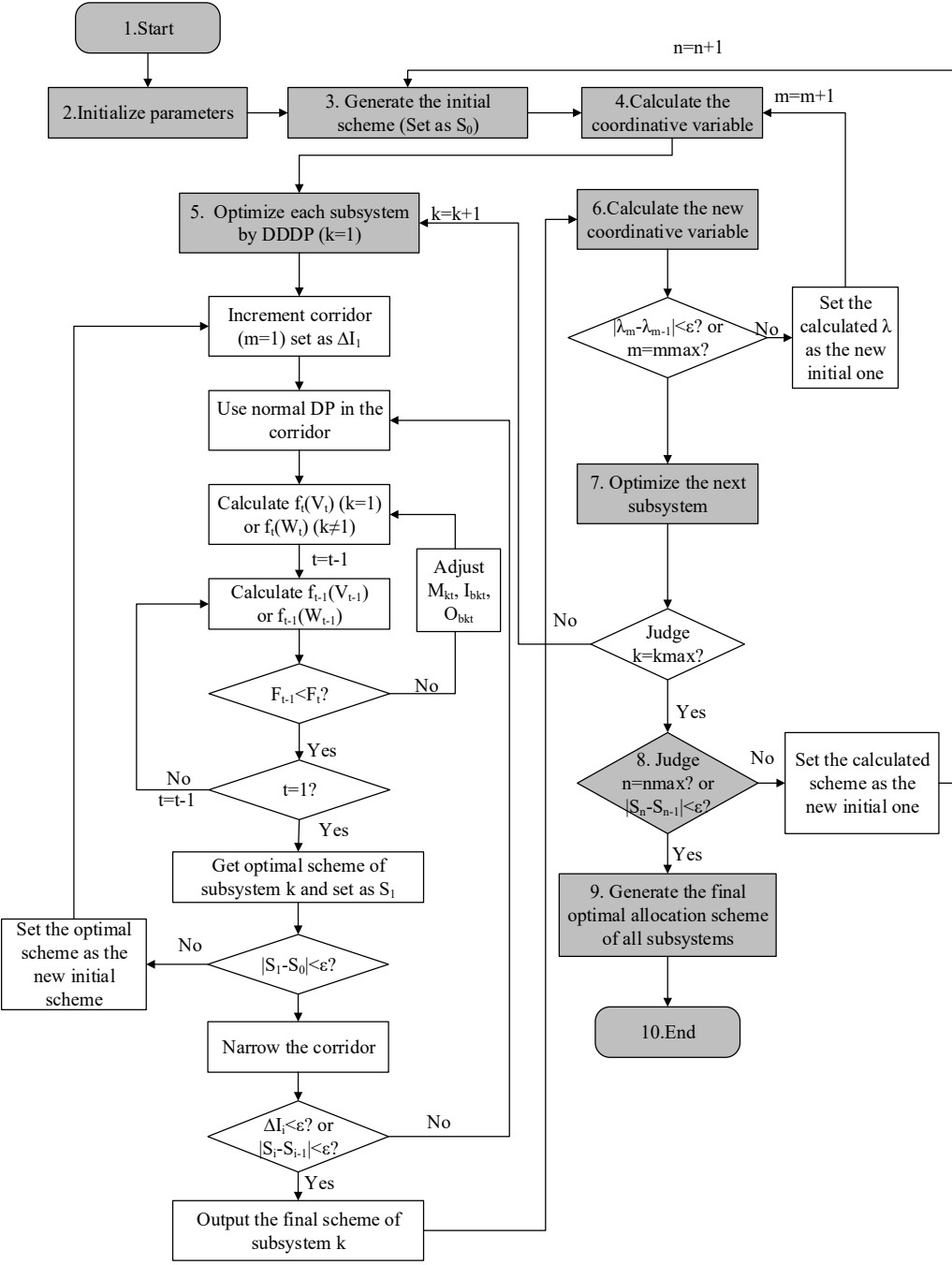

**Figure 3.** The procedure of large system for water resources.

## 4. Case Study

### 4.1. A Brief Description of the Study Area

Lingui and Yongfu County are located to the south west of Guilin City (Figure 4), ranging from 24°42′ N to 25°30′ N in latitude, and 109°42′ E to 110°18′ E in longitude. It is an area with a population of 0.31 million and an annual Gross Domestic Product (GDP) of 2.26 million Renminbi (RMB) in 2013. It has a mild climate with an annual average temperature of 18 °C. The study area has a subtropical monsoon climate with an average rainfall of 1800–2600 mm. The continued acceleration of population growth and the rapid development of the economy presents the region with sever water-related challenges, such as water deficit and sustainability of the riverine ecosystem. With sharp increases

in the population and the economy, the corresponding water demand will be also be rising rapidly (Table 1). Due to the extremely uneven annual distribution of precipitation and runoff, optimizing the water resources of the study area is of utmost necessity.

To take into account the increasing water demands, this study discusses two scenarios. 2020 and 2030 are selected as short-term and long-term years, respectively. The socio-economic water demand is made up of domestic, industrial and agricultural water demand, and these are calculated based on the changing index of population, GDP and irrigation area. Reservoirs are still the dominant water supply projects for solving the water shortage problem in this area. The study area is divided into six towns, and the big Xihe reservoir, with a total storage of $2.69 \times 10^8$ m$^3$, will be constructed in the short-term year to supply water for these towns. The topological structure of the reservoir and the towns is demonstrated in Figure 4b. The Xihe reservoir will supply water for six towns, either by riverbed or channel. The water system in Lingui Town is different from that in the other towns, which are manually interconnected. For the towns upstream of the reservoir and Lingui Town, water is supplied by channel from the reservoir.

**Table 1.** Economic development index and water demands in different planning years.

| Level Year | Population/Million | GDP/Million RMB | Domestic Water Demand/$10^8$ m$^3$ | Industrial Water Demand/$10^8$ m$^3$ |
|---|---|---|---|---|
| 2013 | 0.31 | 2.26 | 0.29 | 0.87 |
| 2020 | 0.51 | 5.33 | 0.59 | 1.67 |
| 2030 | 1.09 | 10.35 | 1.26 | 2.10 |

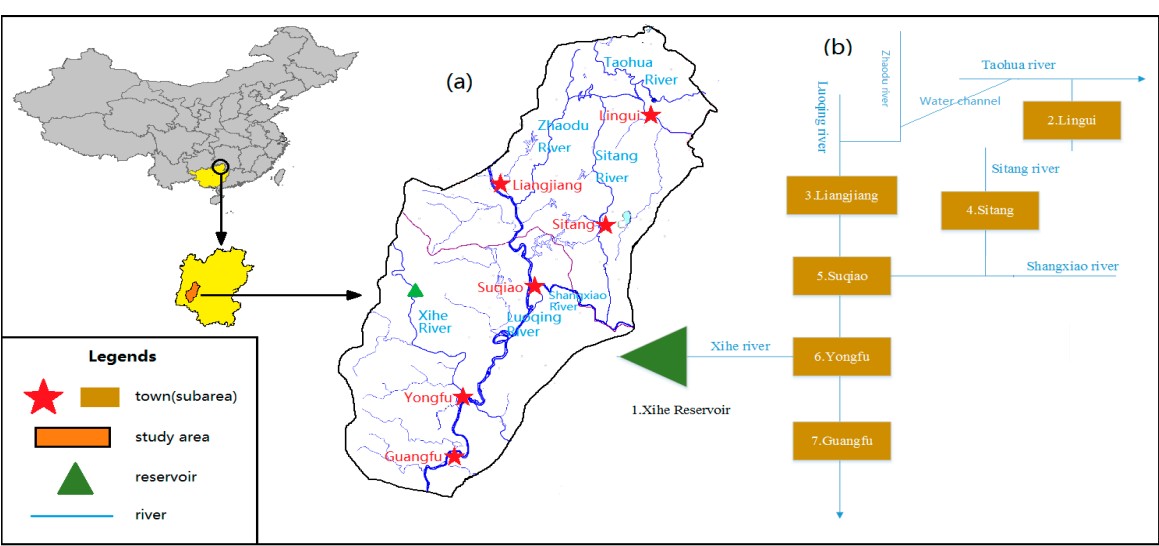

**Figure 4.** Location and water resources condition of the study area. (**a**) Actual river system and sub-area (**b**) the topology structure of rivers and sub-areas.

### 4.2. Data and Methodology

The real-time simulation of the water resources allocation using the system analysis technique should be based on the total volume of incoming water. Here, we collected the natural inflow for one-month periods of the Xihe reservoir from 1958 to 2013 in order to proceed the simulation. The water recession rate was set at 15% of the total water supply of the previous sub-area. The total water demand of each water user at different level years was calculated based on the economic indexes in the statistic yearbook of Guilin city.

In this multi-objective optimization model, there are 4704 decision variables (56 years × 12 months × 7 subsystems). As such, it is a high-dimensional and complicated system that is extremely difficult to handle directly. The decomposition-coordination method and discrete differential dynamic

programming were used to search for optimal solutions. By using the DC method, the objectives of the large system are dispersed to subsystems, and an optimal process is performed for each subsystem taking each objective into full consideration. Obviously, the solution process of several subsystems is much easier to implement than that for a high-dimensional system. In addition, by using the DC method, the summary of the sub-problems is equivalent to the original problem, because the sum of the subsystems is equal to the original large complex system based on the Lagrange multiplier principle. In addition, all objectives can be considered, since the objectives are distributed into different subsystems. Therefore, the aforementioned methods greatly decrease computation time and are more effective because the total calculation amount grows linearly rather than exponentially [7].

To effectively obtain the solution for the case study, the DC method was used to divide the whole water supply system (Figures 1d and 4a,b) into a reservoir and several water supply areas, and their specific expression can be seen in Equation (14). Moreover, the given inflow data are determined as an important input of the whole system. The implementation process of DC and DDDP is presented in Figure 3, and the detailed steps were described in Section 3.3.

### 4.3. Results and Discussion

#### 4.3.1. Analysis of the Optimal Allocation Result

The effects of obtaining an optimal result for the reservoir can be reflected in the trends of both the natural flow and reservoir release over the years. However, to decrease the degree of deviation, the following conditions should be satisfied: (1) the trend of the annual variance of actual water should be identical to that of the natural flow, and (2) the difference between actual reservoir release and natural flow should not be too great. Figure 5 illustrates the changing trend of both natural flow and actual release.

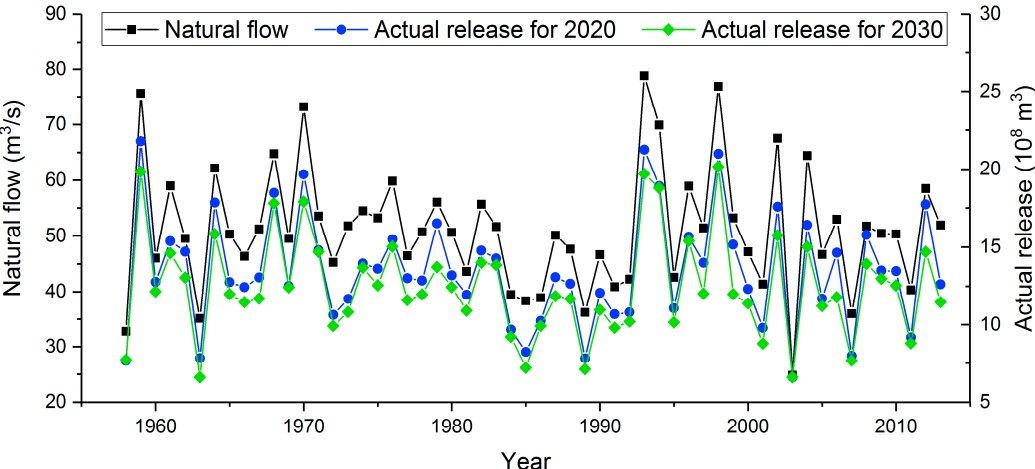

**Figure 5.** Long serial changes of natural flow and actual release.

As can be seen in Figure 5, the long serial change of natural flow and the actual release for both 2020 and 2030 are almost the same. According to Bunn et al. (2002) [39] and Yin et al. (2010) [20], this changing trend between natural flow and reservoir release expressed in Figure 5 indicates that the long serial reservoir release of both scenarios is beneficial for protecting the ecosystem of the river. When it comes to the difference between these two items, there still exists a gap, but it is not enough to deteriorate the ecological environment. The gap occurs because the natural flow is not only used entirely to maintain the riverine ecosystem, but is also used to satisfy socio-economic needs. To further illustrate the trend of natural flow and actual release, we selected three typical yearly runoff series from the long serial runoff data to analyze the annual distribution of the actual release (Figure 6 and Table 2). It can be seen from Figure 6 that the changing trends of the annual distribution of reservoir

release and natural flow are also almost the same. It should be noted that there is no obvious decrease in actual in particularly dry years, because the amount of reservoir release in particularly dry years is equal to the basic flow in both scenarios.

Figure 5 also apparently implies that the amount of actual reservoir release for 2030 for each time period will be a little smaller than that of 2020, and the value of $F_2$ is provided in Table 3. We find that the value of $F_2$ in 2030 is higher than that in 2020 under the same water conditions, apart from particularly dry years, which only suggests the total amount of reservoir release is decreasing, and the trend in years with different levels stays the same (Figure 6 and Tables 2 and 3). This reduction occurred due to the sharp increase in economic index and water demand in the study area (Table 1), and the water release of the reservoir can be reduced as appropriate to supply more water for maintaining the economic development on the condition that the riverine environment is protected. This means that the optimization result for reservoir release can be flexibly adjusted based on both socio-economic water demands and internal water conditions, and the system model construction is applicable for reservoir operation and management, which is beneficial to sustainable development.

**Table 2.** Actual release ($Q$) in different water conditions (unit of release: $10^8$ m$^3$).

| Scenarios | Normal Year | | Moderate Dry Year | | Particular Dry Year | |
|---|---|---|---|---|---|---|
| | Total Release | $F_2$ | Total Release | $F_2$ | Total Release | $F_2$ |
| 2020 | 13.08 | 0.1848 | 10.80 | 0.2384 | 6.59 | 0.2803 |
| 2030 | 11.85 | 0.2214 | 10.21 | 0.2768 | 6.59 | 0.2803 |

**Table 3.** The actual average release ($Q$) of downstream Xihe reservoir in different scenarios.

| Scenario | Optimal Reservoir Release for Each Month ($10^8$ m$^3$) | | | | | | | | | | | | Total | $F_2$ |
|---|---|---|---|---|---|---|---|---|---|---|---|---|---|---|
| | 1 | 2 | 3 | 4 | 5 | 6 | 7 | 8 | 9 | 10 | 11 | 12 | | |
| 2020 | 0.26 | 0.39 | 0.71 | 1.55 | 2.46 | 3.08 | 2.17 | 1.22 | 0.58 | 0.39 | 0.37 | 0.29 | 13.47 | 0.2948 |
| 2030 | 0.23 | 0.35 | 0.61 | 1.41 | 2.36 | 2.89 | 2.04 | 1.13 | 0.54 | 0.33 | 0.32 | 0.24 | 12.46 | 0.3266 |

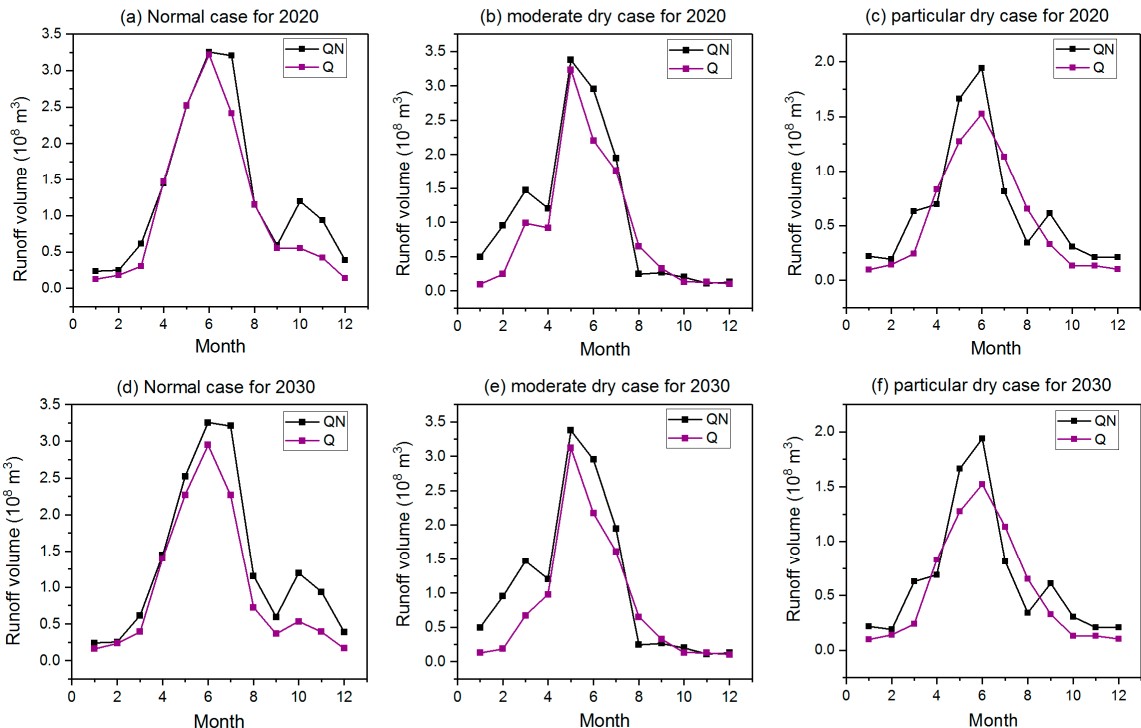

**Figure 6.** The annual distribution of natural flow and actual release.

When it comes to the water allocation scheme, it is clear that the total water supply is on the rise as water demand increases, but the water deficit is also increasing at the same time, with the exception of the normal year case (Figure 7). Although the actual reservoir release is reduced in 2030 compared to 2020 in order to support economic development, there is still an increasing water deficit ratio due to the sharply increasing economic index, resulting in the growth rate of the regional water supply failing to catch up to the growth rate of the water demand. Fortunately, the water deficit ratio, on average, is not enough to restrict economic development.

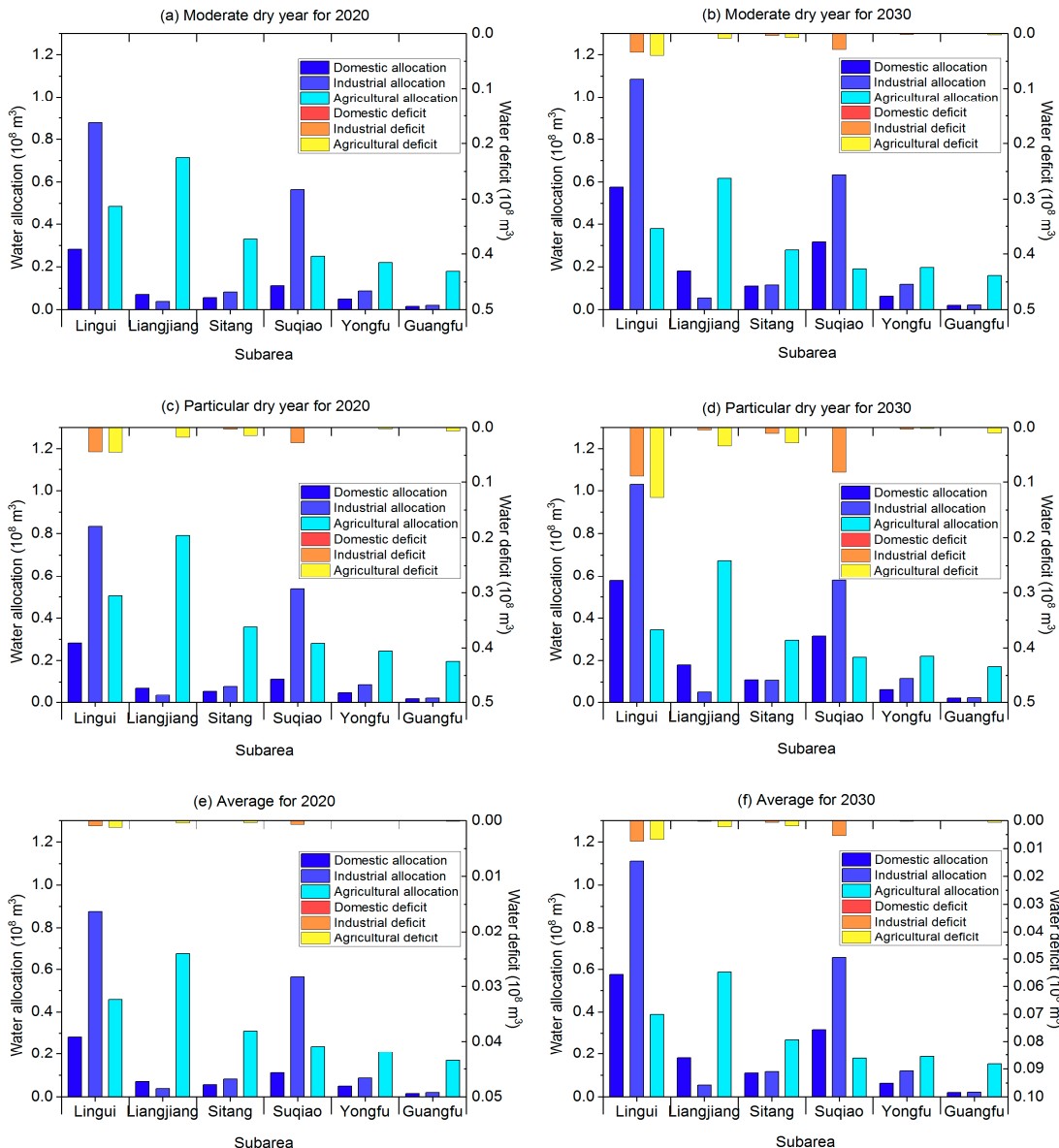

**Figure 7.** Optimal water allocation scheme for different water users under different scenarios.

We also compared the schemes of optimal water allocation for different water users under different level years and hydrological conditions. Figure 7 shows that the optimal allocation for domestic and industrial water supply almost stayed the same under different hydrological conditions in the same level year. However, industrial water supply in particularly dry years is slightly lower than that in moderately dry years in some sub-areas because of its lower priority compared with the domestic water supply. In addition, the study area only has an industrial and agricultural water deficit, and no domestic water deficit, and the difference between these two water deficits is not too big. This is

because the water allocation should conform to a relatively "balanced priority" [26], with high priority being given to domestic water users [4]. Moreover, the water deficit under different hydrological conditions varies since the agricultural water demand is greatly related to hydrological conditions. Fortunately, agricultural water demand is decreasing in order to support the industrial production and people's lives, and its deficit in average and moderately dry years is not too serious.

### 4.3.2. Trend and Sensitivity Analysis

In this research, each subsystem has a direct or indirect mutual relationship, but each sub-area was directly connected to the reservoir, because the reservoir supplies water to each sub-area. Thus, the release of the reservoir would directly affect the water allocation results. The sensitivity analysis of the whole model is mainly embodied by the parameters $F_1$ and $F_2$. $F_1 = 0$ means that water demand can be fully satisfied for each sub-area in each time period, and $F_2 = 0$ implies there is no water alteration of natural flow. Figure 8a,b show the effects of the change of average reservoir release on the value of Lagrange function ($L$), the water supply objective ($F_1$) and the ecological objective ($F_2$) in both scenarios, respectively. There obviously exists a balance point between these two variables, because $F_1$ and $F_2$ have an opposite trend. It can also be observed that the value of the x-coordinate corresponds to the minimum value of the Lagrange function and is exactly the critical point where the value of $F_1$ begins to increase. The critical point and the minimum Lagrange value are on the optimal line that is vertical to the x-axis. This also reveals that the point is exactly the biggest value of the average actual reservoir release, and also the smallest value of $F_2$ under the condition of minimum water deficit ($F_1$) when $L$ achieved the optimal value. This phenomenon implies that it is just the balance point, and that the optimal result of the water allocation scheme is when $L$ reaches its minimum.

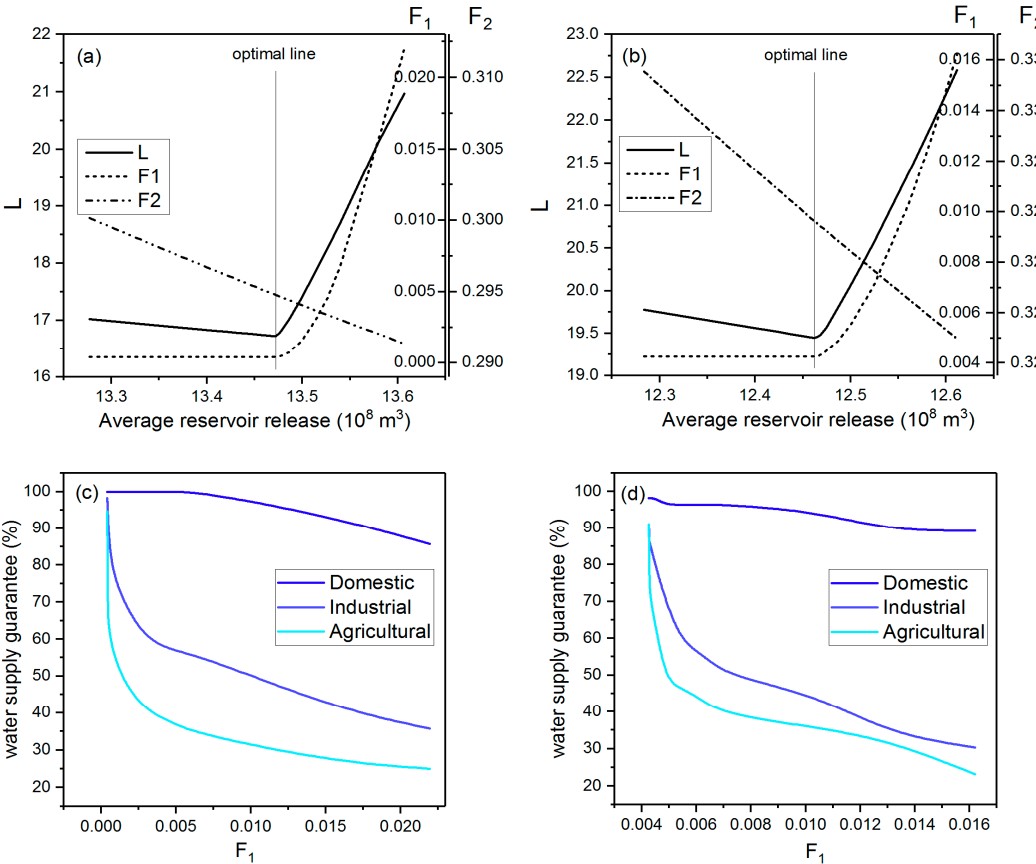

**Figure 8.** Sensitivity analysis of parameters: (**a,c**) 2020; (**b,d**) 2030.

We can also observe from Figure 8 that the changing trend of $F_1$ was much more drastic than $F_2$ when actual reservoir release was greater than the optimal line, meaning that the water supply objective was much more sensitive to the whole system model than the ecological objective. In addition, the high sensitivity of the water supply element is also manifested by the effects on guaranteed water supply for different water users (see Figure 8c,d). Water supply guarantee rate, especially that for industrial and agricultural users, declines dramatically as $F_1$ increases slightly because of their secondary priority. Moreover, the different sensitivities were also reflected in the changing rate of the Lagrange function. When the reservoir release exceeds the optimal line, it means the extra release to protect the riverine environment is at the expense of a reduction in the water supply, and the more sensitive water supply element also made $L$ increase dramatically. Conversely, reservoir release that is less than the optimal line has no effect on the water supply because it only influences the ecological objective, and the changing trend of $L$ is much smoother. Different water demand levels could also influence the value of $L$ and $F_1$, as well as the average reservoir release. The total water demand is increasing as the economy develops, so the layout of water allocation will also be adjusted. Decision-makers should satisfy the socio-economic water demand due to its higher sensitivity, and ensuring water supply only slightly decreases reservoir release and decreases the guarantee rate of ecological water demand.

### 4.3.3. Comparisons with Traditional Allocation Schemes

This study also examines the improvements in the optimal model of a large complex system by compared to the traditional allocation scheme. The traditional model mainly focuses on social benefit, since ecological benefit is not completely maximized. The comparison of water resources allocation between the two models is demonstrated in Figure 9. The results show that the water allocation in the optimized model is greater than in the traditional model, except for the domestic water supply, and this is strongly reflected in the case of particularly dry years. In the traditional model, riverine ecological water demand is usually determined using the Tennant method [4,45]. According to this method, the percentage of the monthly natural runoff is usually set at different values for the flood season and the non-flood season, respectively [46], but it usually fails to consider the interannual variation of the natural runoff. It is also too arbitrary to determine the percentage of the annual average runoff, and reservoir release is therefore not able to play a big role in ecological protection. In this study, ecological concerns are more strongly emphasized in order to bring the reservoir release closer to both the annual and interannual natural runoff than is possible in the traditional model, with lower reservoir release in extremely dry periods resulting in greater water allocation than traditional scheme (Figure 9a,b), because the extra release is used to support the social water supply. In average cases, the percentage of the annual average runoff in the traditional model is usually overestimated because of the abundant water resources of the study area, which also makes the amount allocated using the traditional model less than in the optimal scheme.

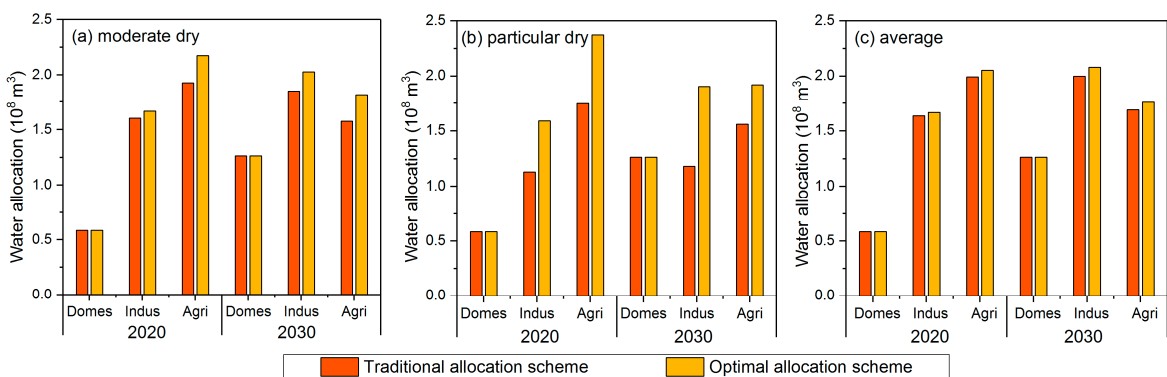

**Figure 9.** The results of water allocation in both traditional and optimal cases.

The advantage of the optimal allocation scheme can be intuitively observed by the value of the objective function of the entire water resources model. Table 4 shows the objective parameters of the system model in both the traditional and the optimal models. It can be seen that the parameters in the optimal model are always smaller than those in the traditional model. This phenomenon not only implies that there would be less alteration of the hydrological regime in the optimal model, but there would also be more water supply and less water deficit in the study area. The optimal model mainly applies the system analysis technique to optimize this large system by introducing the decomposition-coordination method, and the value of $L$ also decreases. Therefore, this method can take both water supply and ecological benefits into full consideration, because the optimization process is implemented in each subsystem, which contain each of the sub-objectives, in contrast with the traditional model.

**Table 4.** The value of objective function between traditional and optimal scenarios.

| Level Year | Parameters | | | | | |
| --- | --- | --- | --- | --- | --- | --- |
| | $L$ | | $F_1$ | | $F_2$ | |
| | Traditional | Optimal | Traditional | Optimal | Traditional | Optimal |
| 2020 | 33.91 | 16.72 | 0.0299 | 0.0004 | 0.5094 | 0.2948 |
| 2030 | 36.67 | 19.45 | 0.0445 | 0.0043 | 0.5094 | 0.3266 |

## 5. Conclusions

This study developed a large complex system of water sources, and the optimal solution of allocation scheme was solved using the system analysis technique. The optimal water resources allocation model takes into account both water supply and environmental needs, and the combination of the decomposition-coordination and discrete differential dynamic programming methods was used to optimize the model efficiently. These two main methods can reduce the dimensions of the complex large system to increase the solution efficiency by distributing the two objectives into different subsystems, and each objective can thus be taken into account by optimizing each subsystem. It can then be integrated into the process of water resources planning and management based on sustainable development. The following conclusions can be drawn based on this study:

(1) There exists a balance between ecological and socio-economic targets, although they can be mutually contradictory. The total optimal allocation result will be obtained if the ecological release of the reservoir is as close as possible to the natural flow while achieving minimal water deficit. In the case study, the trend of ecological release of the reservoir was consistent with the natural flow on both year and multi-year scales, and the water supply to different water users could almost be satisfied in order to support economic development in different level years, which is, to some extent, beneficial to sustainable development.

(2) The water supply element is more sensitive to the water resources system model than the ecological element; thus, the actual release of the reservoir can be decreased slightly to increase the satisfaction of the demand for water in different sectors based on the minimum alteration of natural flow and the maintenance of basic ecological flow. This further indicates that the optimal allocation scheme can be flexibly adjusted as the external condition changes.

(3) The case study demonstrated that the proposed model is able to find the optimal water resources allocation scheme in consideration of sustainable water resources utilization in terms of both water supply and riverine ecological benefit, as compared to traditional model, and it can be applied in other, similar, fields where it is necessary to satisfy two conflicting aspects.

There is much space for the development in this study. For example, uncertainly is usually reflected in the random precipitation in both time and space, as well as the irregularity of water demands in different periods, which can be an important factor in water resources management. In addition, the riverine ecology itself is also a large complex system that includes a large number of

elements. The amount and the distribution of actual release of the reservoir are just one of the most crucial elements in this system, but other subtle elements, such as sedimentation, water quality and aquatic condition, are also important in maintaining ecological health. Therefore, uncertainty analysis, comprehensive ecology, and also study on multi-function reservoirs [47] are future research directions in our study.

**Author Contributions:** Y.T., Z.D. conceived and designed the study with support from all authors. Z.Z. and L.H. collected the data; C.X. and Z.D. developed the water resources system model; Y.T. calculated and analyzed the data and completed the manuscript.

**Funding:** This paper was financially supported by National key research and development program (No.2016YFC0401306) and Guangxi Hydraulic Engineering Project (No. ghzx15039G).

**Acknowledgments:** The authors are grateful to Changjiang Institute of Survey, Planning, Design and Research for providing valuable economic and hydrological data. The authors are also grateful to the insight and views of the reviewers and editors.

**Conflicts of Interest:** The authors declare no conflict of interest.

## Abbreviations

| | |
|---|---|
| *TWD* | Total water demand |
| *TWS* | Total water supply |
| *TWDR* | Total water deficit ratio |
| *Q* | Actual reservoir release |
| *QN* | Natural streamflow |
| *V* | Actual reservoir storage |
| *W* | Water quantity of sub-area |
| *WS* | Water supply only from reservoir |
| *WEVP* | Surface evaporation of reservoir |
| *QB* | Basic ecological flow |
| *WR* | Water recession |
| *WIF* | Intermediate flow |

## Appendix A

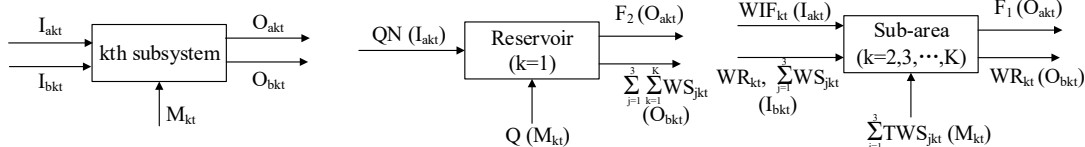

**Figure A1.** The subsystems of the water resources system.

**Table A1.** System model parameters.

| Parameters | Corresponding Meanings |
|---|---|
| $I_{akt}$ | Individual input of subsystem $k$ at $t^{th}$ time step |
| $O_{akt}$ | Individual output of subsystem $k$ at $t^{th}$ time step |
| $I_{bkt}$ | Intermidiate input of subsystem $k$ at $t^{th}$ time step |
| $O_{bkt}$ | Intermidiate output of subsystem $k$ at $t^{th}$ time step |
| $M_{kt}$ | Decision variable of subsystem $k$ at $t^{th}$ time step |

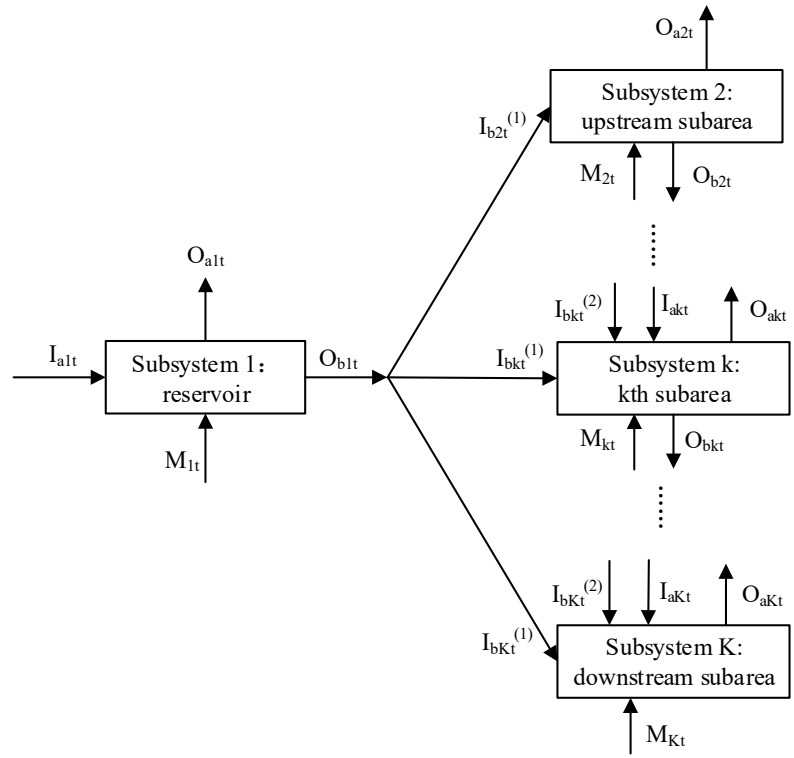

**Figure A2.** The mutual relationship of all subsystems of the water resources system.

**Table A2.** The main parameters in large system model of water resources.

| Subsystem | Parameter for System Modeling | The Corresponding Parameter for Modeling of Water Resources | The Corresponding Meaning of Parameters |
|---|---|---|---|
| Subsystem 1 | $I_{a1t}$ | $QN$ | The natural flow income of reservoir |
| | $O_{a1t}$ | $AAPFD$ | The extent to which actual release to natural flow, the manifestation of the objective function expressed by $F_2$ |
| | $O_{b1t}$ | $\sum\limits_{j=1}^{J}\sum\limits_{k=2}^{K} WS_{jkt}$ | Water supply from the reservoir to each sub-area |
| | $M_{1t}$ | $Q$ | The actual water release of reservoir |
| Subsystem $k$ | $I_{bkt}^{(1)}$ | $\sum\limits_{j=1}^{J} WS_{jkt}$ | Water supply from the reservoir to $k^{\text{th}}$ sub-area |
| | $I_{bkt}^{(2)}$ | $WI_{kt}$ | Water income from upstream sub-area(s), equal to the summary of the water recession (*WR*) of upstream sub-area(s) |
| | $I_{akt}$ | $WIF_{kt}$ | Intermediate water flow between the $(k-1)^{\text{th}}$ and $k^{\text{th}}$ sub-area |
| | $O_{akt}$ | $TWDR_k$ | The total water deficit ratio for $k^{\text{th}}$ subsystem, the manifestation of the objective function expressed by $F_1$ |
| | $O_{bkt}$ | $WR_{kt}$ | Water recession of $k^{\text{th}}$ sub-area |
| | $M_{kt}$ | $\sum\limits_{j=1}^{J} TWS_{jkt}$ | The total water supply for $k^{\text{th}}$ sub-area, including local water, upstream recession and water supply from the reservoir |

Notes: (i) The intermediate input in the system ($I_{bkt}$) denotes different variables in water resources field. For accuracy, the superscript number (1) and (2) is defined to explain the difference. (1) is the water supply from the reservoir to subarea(s) and (2) is water income from upstream sub-area(s). (ii) For convenience, the serial number of each subsystem included the reservoir, and the sub-areas would not be numbered separately. When $k$ only refers to the sub-area in some formulas or equations, its subscript begins with 2.

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
