# Peer review of "An Optimal Allocation Model for Large Complex Water Resources System Considering Water supply and Ecological Needs"

_water, doi:10.3390/w11040843_

Round 1

Reviewer 1 Report

1.      The title has the word mountain but the paper does not reveal the difference between be mountainous and not. If the methodology remains the same for both mountainous and non, please delete the mountainous word.

2.      The paper is about optimisation. However, contrary to what the paper transmits, optimisation cannot give one solution. It basically gives non-dominant possibilities.

a.       In water systems dominant solutions may prove to be better because of high complexity. Even resilient and sustainably efficient water systems should be dominant.

3.      One of the objective functions has to do with ecology, which is presented as a water quantity function. However, other major issues of ecological disasters are water quality, river meandering schemes and sedimentations which are absent from the paper. This should be made clearer.

4.      There is also the idea of ecological flow (L175). However, ecological function is also about a kind of water flow. Their difference is a bit confusing.

5.      Reservoirs are crucial for energy production, however, the paper is totally mute about energy. In reality the idea of reservoirs links water, energy and food (WEF). Again paper lacks such analysis. This means that the utility of the paper is minimum. For a WEF analysis, please see Haie (2016, IJWRD at  https://www.tandfonline.com/doi/full/10.1080/07900627.2015.1070091 ).

6.      L181: why water supply should be less than water demand?

Author Response

Thank you for your suggestion to my paper and all your suggestions are of great significance, which has important guiding significance for my thesis writing and research working.

Point 1: The title has the word mountain but the paper does not reveal the difference between be mountainous and not. If the methodology remains the same for both mountainous and non, please delete the mountainous word.

Response 1: The paper mainly discusses the reservoir construction and its impact on both social water supply and ecology. Reservoir release is indispensable factor in the model and I have put it in the objective function. Reservoirs are usually constructed in mountainous areas, which is different from plain areas with river networks that has no reservoirs, of course there’s no concept of “reservoir release” in plain areas. I have supplemented this in my manuscript in section 2.1.

Point 2: The paper is about optimization. However, contrary to what the paper transmits, optimization cannot give one solution. It basically gives non-dominant possibilities. In water systems dominant solutions may prove to be better because of high complexity. Even resilient and sustainably efficient water systems should be dominant.

Response 2: Thank you very much for your suggestion and it is of great significance. Multi-objective optimization really cannot give the only optimal solution but can give many possible optimal solution set, which is also called non-dominant Pareto-optimal front. One possible solution is usually enough to give a reference to water resources management but dominant solutions may be better than non-dominant ones, which is needed to be better improved. In my further research I will pay more attention to this point to improve my scientific research level.

Point 3: One of the objective functions has to do with ecology, which is presented as a water quantity function. However, other major issues of ecological disasters are water quality, river meandering schemes and sedimentations which are absent from the paper. This should be made clearer.

Response 3: This suggestion is also of great of significance to me. This paper mainly discuss the water quantity function to ecology. Though water quantity is of great importance in river’s health, other functions are of also significance. Due to the time constraints and lack of relevant data, I would present it as my future research and I added it in the last paragraph of the Conclusion section.

Point 4: There is also the idea of ecological flow (L175). However, ecological function is also about a kind of water flow. Their difference is a bit confusing.

Response 4: This is basic ecological flow. It is the minimum flow that is used to maintain the fundamental function of a river. If the actual reservoir release is below this, the river will suffer irreversible damage. The actual release contains the basic ecological flow and it is one of the constraint of the model. 

Point 5: Reservoirs are crucial for energy production, however, the paper is totally mute about energy. In reality the idea of reservoirs links water, energy and food (WEF). Again paper lacks such analysis. This means that the utility of the paper is minimum.

Response 5: Thank you very much for this suggestion and it is also of great significance. This paper mainly discuss the water scarcity in a region as the total water demand is increasing sharply. Without reservoir there will be water scarcity. Of course, the function of the reservoir is more than just water supply, it also has other functions like energy. In the future research the joint reservoir operation with energy, water supply and fold control can be deeply proceeded. I have added it as my reference in the Conclusion section as the future direction of the paper.

Point 6: L181: why water supply should be less than water demand?

Response 6: The amount of water supply is based on the water demand in a region. Water supply and water demand is clarified by many water users such as domestic, industrial and agricultural, and water supply for different water users is usually unable to convert to each other because of the different water facilities. If more water is supplied it will cause a waste of water.

Thank you for your suggestions again, and I hope I can learn more from you.

Reviewer 2 Report

The topic proposed is a very actual issue in water resources management. I proposed a reflection regarding the time step used in the model, and about the use of priority method.

1)The monthly time step may not highlight significant shorter critical periods for water supply, and in particular for ecological needs.

2)The use of priority in water allocation is very penalizing for low priority uses, it is possible to study a balanced priority approach (see remark row 426).

In the introduction you can propose your opinion about these points.

Some others remarks:

·         Row 58: about evaluation of excessive exploitation there are other references in the last years, for example: Tiuan et al "Influence of Three Gorges Dam on Downstream Low Flow" Water, 2019; Hung Vuong Phama et al. "Alteration of freshwater ecosystem services under global change-Po river- Red river" Science of the Total Environment, 2019; Casadei et al. " Sustainability of Water Withdrawals in the Tiber River Basin (Central Italy)" Sustainability, 2018; Amador Durán-Sánchez et al. "Sustainable Water Resources Management-A Bibliometric Overview" Water, 2018.

·         Row 143: equation (1), water users are limited to 3.

·         Row 149: Average annual natural flow?

·         Row 163: Equation 3, evaporation from reservoir surface is not evaluated.

·         Row 184: Sorry, what is the meaning of water supply engineering.

·         Row 329: A brief description of rainfall and temperature in the study area would be useful to better understand results

·         Row 332: GDP and RMB, I suppose  Gross Domestic Product and Renminbi

·         Row 384: References Bunn and Yin must be cited according to journal guidelines.

·         Rows 423-425: This part on industrial water supply is not clear.

·         Row 426: The priority approach is very penalizing for agricultural water demand and for low priority use generally. There is also a "balanced priority", see water 2016 (8-12), "Integrated Water Resources Management in a Lake System: A Case Study in Central Italy".

·         Rows 475-476: there is also a modified Tennant method, see Water Resources Management 2014 (28) "A New Modified Tennant Method with Spatial-Temporal Variability"

Row 484: than optimal

Author Response

Thank you for your suggestion to my paper and all your suggestions are of great significance, which has important guiding significance for my thesis writing and research working.

Point 1: The monthly time step may not highlight significant shorter critical periods for water supply, and in particular for ecological needs.

Response 1: Thank you for your suggestion and it offers an important reference to my research. This paper mainly discuss the effect of water quantity to water ecology. Although monthly time step can reflect the water supply and ecological process, the shorter time step is more likely to reveal the process of both water supply and ecology. However, it is a little bit difficult because of the difficulty to collect the daily even hourly datasets, and it’s going to be a lot of work. Daily time step is usually used in flood control in flood seasons and ecological needs considering flow velocity. These can be in my future research.

Point 2: The use of priority in water allocation is very penalizing for low priority uses, it is possible to study a balanced priority approach.

Response 2: This suggestion is also of importance to my research. Balance priority may be better to balance different water users. In the current water resources management, if it is not an agricultural-dominated region, domestic water users is generally of higher priority compared with industrial and agricultural users. (See Water 2016, 8, 364; doi: 10.3390/w8090364). I have revised the constraint listed in section 2.3 (2), deleted “agricultural user is the lowest priority” and I also recalculated the water allocation scheme in the result section.

For above two points, we have added it in my introduction section in my manuscript and marked it as red (started with L70 for point 1 and L79 for point 2).

Point 3: about evaluation of excessive exploitation there are other references in the last years, for example: Tiuan et al "Influence of Three Gorges Dam on Downstream Low Flow" Water, 2019; Hung Vuong Phama et al. "Alteration of freshwater ecosystem services under global change-Po river- Red river" Science of the Total Environment, 2019; Casadei et al. " Sustainability of Water Withdrawals in the Tiber River Basin (Central Italy)" Sustainability, 2018; Amador Durán-Sánchez et al. "Sustainable Water Resources Management-A Bibliometric Overview" Water, 2018.

Response 3: I have added these as my reference in my manuscript.

Point 4: L158: equation (1), water users are limited to 3.

Response 4: 3 is three different water users in this paper, refers to domestic, industrial and agricultural. To be general, we have revised the equation by changing 3 to J, and subsequent similar formulas have also been modified.

Point 5: L164: Average annual natural flow?

Response 5: We have modified as required.

Point 6: Equation 3, evaporation from reservoir surface is not evaluated.

Response 6: We have modified the equation with considering the surface evaporation of reservoir.

Point 7: L200: Sorry, what is the meaning of water supply engineering?

Response 7: We have modified as “water supply project”. It is water project that is used to water supply.

Point 8: A brief description of rainfall and temperature in the study area would be useful to better understand results

Response 8: We have supplemented the brief description as required. It is on L346~348.

Point 9: GDP and RMB, I suppose Gross Domestic Product and Renminbi

Response 9: Yes, GDP denotes Gross Domestic Product and RMB denotes Renminbi. I have denoted in the paper.

Point 10: References Bunn and Yin must be cited according to journal guidelines.

Response 10: We have cited as required.

Point 11: L439-441: This part on industrial water supply is not clear.

Response 11: Writing mistake. The latter “industrial” should be “domestic” The latter part of the whole sentence has modified as “… because of its lower priority compared with domestic water supply”.

Point 12: The priority approach is very penalizing for agricultural water demand and for low priority use generally. There is also a "balanced priority", see water 2016 (8-12), "Integrated Water Resources Management in a Lake System: A Case Study in Central Italy".

Response 12: L441-442 If agricultural water user is the lowest priority, there will be penalizing for agricultural. So I deleted the “lowest priority” and “apparently agricultural water deficit” word to balance priority, and we have recalculated the allocation results to achieve “balanced priority” and added the reference you provided.

Point 13: L493 Rows 475-476: there is also a modified Tennant method, see Water Resources Management 2014 (28) "A New Modified Tennant Method with Spatial-Temporal Variability"

Response 13: Thank you very much for giving the reference about a new modified Tennant method to calculate the ecological flow. It gives me an important reference to my current and future research about it. The proposed method in this study is better than traditional Tennant method but the modified Tennant method may prove to be better than this study. I’ll try to compare it with my current study in my future work as you suggest to improve my research skills.

Point 14: L502: than optimal

Response 14: Missing a space character. I have modified.

Thank you for your suggestions again, and I hope I can learn more from you.

Reviewer 3 Report

The paper entitled "An optimal allocation model for large complex water resources system in mountainous areas considering water supply and ecological needs" is a paper as we like them : well written, quality references, the method is clear and correct, the results are clearly presented and discussed, and the conclusion is supported by the results. Congratulations to the authors !

The subject of the paper also addresses an uprising issue for reservoir managers and offers interesting advances. 

The paper has many acronyms whose choices are not always obvious, I would then have two main suggestions for the authors :

- add an acronym list section between the keywords and introduction section

- better link the acronym to their meanings and to those traditionally used by the scientific community.

For example :

l.146 : acronym for year : y rather than t

l.147 : TWS for total water supply rather than WSP (what does that stand for ?) or TWD for total water demand

l.150 : the scientific unit for flow/runoff/discharge is Q not WN

Please do check all acronyms in the paper and make sure that they are connected to their meaning, adapt and correct them throughout the manuscript.

Once these really minor changes will be applied to the manuscript I would definitely recommend the publication of this paper.

Author Response

Thank you for your suggestion to my paper and all your suggestions are of great significance, which has important guiding significance for my thesis writing and research working.

 Point 1: add an acronym list section between the keywords and introduction section

Response 1: I have added the acronym list as required.

Point 2: better link the acronym to their meanings and to those traditionally used by the scientific community.

Response 2: I have modified all the acronym and it’s connect to their meaning.